published: 30 August 2021

# Construction and Validation of a Brief Pandemic Fatigue Scale in the Context of the Coronavirus-19 Public Health Crisis

Esther Cuadrado [1,2]*, Miguel A. Maldonado [1,2]*, Carmen Tabernero [1,3,4], Alicia Arenas [1,5], Rosario Castillo-Mayén [1,2] and Bárbara Luque [1,2]

[1]Maimonides Biomedical Research Institute of Cordoba (IMIBIC), Cordoba, Spain, [2]Department of Psychology, University of Cordoba, Cordoba, Spain, [3]Department of Social Psychology, University of Salamanca, Salamanca, Spain, [4]Instituto de Neurociencias de Castilla y León (INCyL), Salamanca, Spain, [5]Department of Social Psychology, University of Seville, Seville, Spain

**Objectives:** The chronic restrictions to mitigate the new SARS-CoV-2 virus may result in pandemic fatigue. This study set out to develop a short, reliable, valid, and gender-invariant instrument—the Pandemic Fatigue Scale (PFS).

**Methods:** In the first phase, 300 students responded to a pilot questionnaire that allowed the reduction and refinement of the items. In the second phase, the validity, reliability, and invariance of the scale were explored among a sample of 596 participants.

**Results:** Factor exploratory and confirmatory analyses confirmed a robust adjustment for the bifactorial structure that explained 79,36% of the variance. The two factors identified were 1) people's demotivation in continuing to follow the recommended protective behaviors (*neglect*) and 2) people's boredom regarding the pandemic-related information (*boredom*). The pattern of relations between the Pandemic Fatigue Scale and other variables—find through correlation, mediation, and path analyses—and the gender differences—find in the ANOVA analyses—provided strong evidence of the construct validity. Moreover, the PFS was shown to be invariant regarding gender in a multigroup factor confirmatory analysis.

**Conclusion:** The instrument can be of utility for professionals and researchers to assess pandemic fatigue, a variable that can affect the adoption of protective measure to avoid catching and spreading the virus.

Keywords: COVID-19, pandemic fatigue, scale development, scale validation, protective behavior

**Edited by:**
Franco Mascayano,
Columbia University Irving Medical
Center, United States

**Reviewed by:**
Hiram Ortega,
National Institute of Psychiatry Ramon
de la Fuente Muñiz, Mexico

**\*Correspondence:**
Esther Cuadrado
esther.cuadrado@uco.es
Miguel A. Maldonado
z62mahem@uco.es

This Original Article is part of the IJPH
Special Issue "The impact of the
COVID-19 pandemic on mental health"

**Citation:**
Cuadrado E, Maldonado MA,
Tabernero C, Arenas A,
Castillo-Mayén R and Luque B (2021)
Construction and Validation of a Brief
Pandemic Fatigue Scale in the Context
of the Coronavirus-19 Public
Health Crisis.
Int J Public Health 66:1604260.

## INTRODUCTION

The emergence of Coronavirus-19 (COVID-19) has resulted in the confinement of most people worldwide, homeschooling, working from home, new social distance rules, use of face masks, curfews, perimeter closures of cities or larger areas within countries, and other stricter restrictions to mitigate the effect of the disease [1]. Most of these substantial changes became chronic, in many cases accompanying us for more than a year. Furthermore, throughout this long time, people have been continuously exposed to information overload related to COVID-19, with daily and repeated information on the number of infections, deaths, changes in measures, opening and closing of areas, relaxation and tightening of restrictions, etc.

This resulting constant state of alert and uncertainty has become emotionally exhausting for many individuals [2, 3]. The large and repetitive implementation of invasive measure to mitigate COVID-19 consequences, as well as the overexposition to negative pandemic-related information, has supposed a natural psychological response of pandemic fatigue (PF) on individuals [4].

Many individuals have responded to the chronic nature of the public health crisis with this psychological response characterized by a general negative attitude and demotivation regarding the adoption of the recommended COVID-19 protective behaviors (PB) and a sort of boredom regarding COVID-19 pandemic-related information [4–6]. This PF is a relevant problem because it is related to a diminution of the adoption of individual PB [3, 7]; in turn, a decrease in the adoption of PB can lead to a new increase in cases and a prolongation of the pandemic situation [2, 7, 8].

Exploration of the PF situation of individuals and societies as well as protective and risk factors of PF and potential consequences seems to be of relevance. Nonetheless, as far as we know, there is no existing PF scale. The main aim of this study is to develop and validate a reliable PF instrument. The construct validity and reliability of a PF Scale (PFS) will be tested, and the relationship between the scale and its dimensions with other variables will be evaluated to test the external validity of the scale. Moreover, the invariance across genders will be explored, as well as the potential gender differences regarding PF and its dimensions.

## Pandemic Fatigue: Construct, Measurement, and Related Variables

PF is a term mostly used by the general population, and it is described by the World Health Organization (WHO) [4] as demotivation due to the chronification of the pandemic situation, which negatively affects the protective behavior that people adopt to prevent the propagation and contagion of COVID-19. Although it is a relatively widespread and well-known concept, as far as we know, there are no validated instruments for its measurement. This lack of previous instruments means that we approached the construction of the scale and proposed its potential sub-dimensions without incorporating previous tools to develop the items, so they were thus theoretically built. The definitions given by a number of international institutions and authors [4–6] suggest that this construct could be built on the basis of two different dimensions: demotivation in continuing to follow the recommended PB—that is the primary definition of PF stated by the WHO and other authors [4, 5, 9]—and boredom regarding overexposure to pandemic-related information.

The first dimension proposed is clear, as it represents the definition of PF made by several authors and the WHO [4, 5, 9]. Nevertheless, the second potential dimension proposed for PF—boredom regarding overexposure to pandemic-related information—requires more attention and the description of the conceptual framework. First, mental health is negatively affected by the experience of strict measures and overexposure

to negative information in the media about the pandemic, two aspects that exacerbate the risk of developing post-traumatic stress disorder [10]. It has thus been shown that disclosure of generic and detailed information has the power to increase risk perception and panic in pandemic situations [11]. Moreover, information overload through social media during the pandemic has been shown to heighten social media fatigue [12]. Some authors also refer to messaging fatigue related to the pandemic as fatigue due to chronic and long-term exposure to repeated messages regarding pandemic-related information [13]. Boredom regarding overexposure to pandemic-related information thus seems to be a plausible dimension of PF. Consequently, PF can be defined as a negative attitude toward PB related to the pandemic situation which results from feelings of emotional exhaustion associated with the chronification of COVID-19 restrictions and with overexposure to COVID-19 related information [4–6]. To construct a psychometrically sound measure, the PFS was conceptualized with these two expected dimensions: 1) demotivation in continuing to follow the recommended PB (called neglect) and 2) boredom regarding the pandemic-related information (called boredom). Thus, our first hypothesis (H1) is that the PFS will present a bifactorial structure corresponding to those two dimensions.

Regarding external validity, the relation between these two dimensions and the general PFS with different constructs potentially related to them was explored. In this sense, personal values aligned with conservation, which are oriented both toward the collective interests and toward protection [14], have been related with normative behavior related to COVID-19 prevention [15]. In the first phase of confinement due to the COVID-19 pandemic, individuals oriented toward conservation—who give high relevance to both security and conformity—adopted more protective and normative behaviors [15]. Thus, considering that both conformity and security orientation embody social values of conservation that orient individuals to self-protection [14], we hypothesize that individuals who hold high security and conformity values will show lower PF, and above all a lower demotivation to follow PB as a consequence of the prolonged pandemic situation (meaning a lower level of the neglect dimension of PF), which is the dimension of the PFS that is most related to security (H2).

Evidently, considering the construct of PF, its motivational component, and thus its inherent relation to behavior, the PFS and its two dimensions are expected to be related with both intention to behave in a protective way related to the pandemic and with actual PB, as other authors have suggested [3, 7]. Thus, we hypothesize that people with a high level of PF (in both hypothesized dimensions, boredom and neglect) will show lower levels of PB intentions (H3) and will behave in a lower protective way (H4).

## The Pandemic Fatigue Construct and the Theory of Planned Behavior

The theory of planned behavior (TPB; 11–13), which has been demonstrated to be especially relevant and valued in predicting health-PB [16–18], explains why sometimes people hold an

attitude but do not behave in the consequent way: because other variables also influence behavior and thereby can blur the attitude–behavior link. In short, the TPB [16, 19], explains the motivational influences on behavior by proposing the mediating role of behavioral intention in the relationship of different other unanticipated events or factors—attitudes, social norms, and behavioral control—with behavior. Thus, it posits that behavioral intention is the direct predictor or immediate antecedent of behavior, but that a variety of unanticipated events may prevent individuals from turning their intentions into action.

Therefore, framed by the TPB, we expect that the intention to adopt PB related to the pandemic situation could mediate the relation of both proposed dimensions of PF with the PB individuals carry out. Moreover, the perception that individuals have about whether other individuals of significance for them (friends, family, people with whom they usually interact) engage in these kinds of PB related to the pandemic situation (meaning the perceived social norms) could also be a relevant predictor of their intention to adopt PB, acting as a sort of pressure to behave in a determined way. Finally, the perception that PB related to the pandemic situation are difficult to follow (meaning the perceived behavioral control) could also predict both the behavioral intention to behave in a protective way and the actual PB. In this sense, we propose that the behavioral intention to adopt PB related to COVID-19 prevention will mediate the relations of both dimensions of PF—neglect (H5a) and boredom (H5b)—with COVID-19 PB; in the same way, behavioral intention will mediate the relations of (H5c) social norm perceptions about PB and (H5d) perceived behavioral control with COVID-19 PB (See the predictive model in **Supplementary Figure 1**).

## A Gender Perspective

Various entities have described the differential impact of COVID-19 on men and women, pointing out inequities regarding gender. The WHO [20] has emphasized the differential gender impact of the pandemic and the need to considering COVID-19 research with a gender perspective. Thus, to provide investigators and other professionals with a valid and reliable gender-invariant instrument to measure PF seems to be of relevance. For this purpose, this study will focus on the analysis of the invariance of the PFS structure.

Previous studies have shown that men and women present a different psychological adjustment to extreme situations such as pandemic confinement [21]. Thus, analyzing the differential gender impact on psychological adjustment to the pandemic situation seems to be essential, as is studying the gender differences in PF in the face of the different phases of the pandemic. Other studies have found that women usually adopt more PB oriented to preventing SARS-CoV-2 transmission [22–24], probably because women usually assign more relevance to health issues than do men [25]. In this sense, we expect that women will present lower levels of the neglect dimension of PF. The exhaustion due to the prolonged crisis and its associated reiterative, chronic, and strict restrictions and overexposure to negative information related to the pandemic

will probably influence men and women similarly regarding the dimension of boredom; nevertheless, because women are more focused on health than are men, this fatigue will probably entail a lower demotivation to follow the recommended PB to prevent SARS-CoV-2 transmission on women compared to men (H6).

## METHOD

### Procedure

The study was conducted in two different phases. In the first phase, a questionnaire was administered to a student pilot sample, to reduce and refine the items. In the second phase, a questionnaire was administered to a larger sample to assess the validity and reliability of the scale. In both phases, conducted before the 2020 Christmas holidays, informed consent was obtained. The study was conducted in accordance with the Declaration of Helsinki. Before giving their consent, participants were informed about the study objectives and the voluntary nature of their participation, that their anonymity was ensured, and that they could withdraw from the study whenever they wanted.

On December 1, 2020, for phase 1, by using a convenience sampling method, the researchers requested that their university students (pilot sample) complete an online questionnaire containing sociodemographic questions as well as the six items of the PFS. Moreover, students were asked to disseminate a message among their social networks in which both further diffusion on social networks and response to the phase 2 questionnaire were requested.

On December 15, 2020, for phase 2, by using a combination of convenience and snowball sampling methods, pilot sample participants and researchers shared a link to the second questionnaire. The sample was obtained between December 15 and December 22, 2020.

### Participants

The questionnaires were completed by 300 students (75.3% of women; age range = [18, 24], $M = 19.21$, $sd = 1.65$) and 596 participants (64.4% of women; age range = [18, 80], $M = 42.62$, $sd = 13.90$) in the first and second phase, respectively (See sociodemographic data in **Supplementary Table 1**).

### Measurements
#### The Pandemic Fatigue Scale

A brief six-item questionnaire (see **Table 1**) was created to analyze the level of PF, understood as a natural response to the large and repetitive implementation of invasive measures to counteract the prolonged public health crisis, as well as to the information overload regarding the pandemic situation. These responses are expressed as a general demotivation to follow recommended PB and boredom regarding COVID-19 pandemic-related information [4]. The items were divided into two pools: 1) three items related to demotivation in continuing to follow the recommended PB (called *neglect*) and 2) three items related to boredom regarding the pandemic-related information (called *boredom*). Participants responded to these items using a Likert-type scale from 1 ("strongly disagree") to 7 ("strongly agree").

**TABLE 1 |** Results of the Exploratory Factorial Analysis of the Pandemic Fatigue Scale: Factor Loadings, Reliability Estimates, and Percentage of Explained Variance. Study Attitudes, behaviors, and psychological health in time of pandemic, Spain, 2021.

| Items | Highest loading for each one of the two factors | |
| --- | --- | --- |
|  | F1 | F2 |
| 1. I am fed up with the COVID topic being talked about so much in all the media | — | 0.872 |
| 2. When someone starts talking about COVID, I am disinterested | — | 0.779 |
| 3. I do not want to hear more about the COVID issue | — | 0.877 |
| 4. I am already so tired of the COVID issue that I am not as careful as I was at the beginning | 0.888 | — |
| 5. So much time immersed in the pandemic discourages me from adopting protection measures against COVID | 0.883 | — |
| 6. I am already so fed up with COVID that I no longer adopt certain protection measures that I would have taken before to avoid becoming infected | 0.849 | — |
| Scale reliability estimates | | |
| Cronbach's alpha values | 0.85 | 0.89 |
| Percentage of explained variance | 59.28 | 20.08 |

## Security and Conformity Values of Social Orientation Values

To measure to what extent individuals had personal values oriented toward self-protection, three items of the security factor and four items of the conformity factor of the Portrait Values Questionnaire [14] were used. For each of the seven items, participants indicated the degree to which they self-identified with a description of a person on a 7-point Likert-type scale (from 1 = "not like me at all" to 7 = "very much like to me"). The variables security and conformity were high for both the pilot (α = 0.76 and 0.85, respectively) and the general sample (α = 0.79 and 0.78, respectively).

## Perceived Social Norms About Protective Behaviors Related to Coronavirus-19 Transmission

To measure to what extent participants perceived that valued others think that PB related to COVID-19 prevention are necessary (social norms of thinking) and adopt those PB (social norms of actions), 11 items (See **Supplementary Table 2a**) were created ad hoc from the social norms construct of the TPB [16, 26] and adapted to the pandemic context. Participants responded on a 7-point Likert scale to what extent they perceived that the people with whom they usually interact believe that PB oriented to prevent COVID-19 transmission were necessary and adopt those PB (higher level of the measure indicated that participants perceived that the people with whom they usually interact 1) think that the PB are exaggerated and 2) do not adopt them frequently). The reliability of the scale was high (α = 0.84 for the pilot sample and α = 0.88 for the general sample).

## Perceived Behavioral Control About Protective Behaviors Related to Coronavirus Transmission

To measure to what extent participants perceived that the PB to prevent COVID-19 transmission were easy or difficult to carry out, nine ad-hoc items (See **Supplementary Table 2b**) were created on the basis of the perceived behavioral control construct of the TPB [16, 26] and adapted to the pandemic context. The reliability of the scale was high for both the pilot (α = 0.87) and the general (α = 0.87) samples.

## Intention to Adopt Protective Behaviors Related to Coronavirus Transmission

To measure participants' intention to adopt PB to prevent COVID-19 transmission, eight ad-hoc items (See **Supplementary Table 2c**) were created on the basis of the perceived behavioral control construct of the TPB [16, 26]. Participants responded to the items by indicating on a 7-point Likert scale to what extent they had the intention to perform different PB to prevent virus transmission in their family gatherings (lunches and dinners) during the next Christmas (remember that the questionnaire was completed only a few days before Christmas). The reliability of the scale was high for both the pilot (α = 0.81) and the general (α = 0.82) samples.

## Protective Behaviors

To measure to what extent participants adopted PB to prevent COVID-19 transmission, a nine-item ad-hoc frequency scale was created (See **Supplementary Table 2d**). Participants responded on a 7-point Likert scale how often they adopted the proposed PB against COVID-19. The reliability of the scale was high for both the pilot (α = 0.89) and the general (α = 0.94) samples.

## Statistical Analyses

In phase 1 (pilot sample), in which the main objective was to reduce and refine the PFS, items were subjected to prior data checks to explore their suitability for inclusion in the subsequent analysis (missing values, floor and ceiling effect, inter-items and items-total correlations, and reduction of the value of Cronbach's alpha). Then an exploratory factor analysis (EFA) with varimax rotation was performed to identify subscales within the item sets. The appropriateness of using factor analysis was assessed using Bartlett's test of sphericity (BTS; 24) and the Kaiser-Meyer-Olkin statistic (KMO; 24).

In the second phase, with a different and balanced sample [27, 28], a confirmatory factor analysis (CFA) was performed, using Amos.26. To interpret the goodness of fit of the different indices, we used the rules of thumb recommended by Schermelleh-Engel et al. [29]. Moreover, a multigroup CFA analysis was performed to explore the invariance of the scale regarding gender. Then, to explore the external validity of the final scale, correlations

between the PFS and other related variables were evaluated, as well as some mediation analyses and a path analysis. To evaluate the mediation hypotheses, mediation analyses were performed by using model 4 of the Process for SPSS macro [30] with a confidence interval of 95% and 10,000 bootstrap resamples. To confirm the hypothesized predictive model of PB related to COVID-19 prevention, a path analysis was performed with AMOS.26. The adjustment and goodness of fit of the model were interpreted with the rules of thumb recommended by Schermelleh-Engel et al. [29]. The invariance of the scale was tested through multigroup CFA analysis. The invariance was explored by comparing the $\chi^2$ of the unconstrained and fully constrained model. Moreover, the differences by gender in the PFS were observed through ANOVA.

# RESULTS

## Phase 1: Reducing and Refining the Items
### Floor and Ceiling Effects
No items had more than 10% of missing data. Item 6 ("I am already so fed up with COVID-19 that I no longer adopt certain protective measures that I would have taken before to avoid becoming infected") showed a floor effect, with 53% of the respondents selecting the lowest response. Despite this result, we retained the item in accordance with previous literature showing that despite the negative effects that PF can produce, the population, in general, continues to maintain the measures of care and protection against COVID-19 [6, 31]. Thus, the 53% of responses choosing the most extreme scale option reflects that people continue to observe protection measures to avoid the risk of infection even when exhausted with the pandemic situation.

### Reliability Analysis and Correlations
The reliability level of the general scale ($\alpha = 0.86$) and the neglect and the boredom factors (**Table 1**) were all elevated. No item showed a poor correlation ($r < 0.20$) with half of the other items in each factor, a low item-total correlation ($r < 0.25$), nor decreased Cronbach's alpha if removed. Consequently, no item was eliminated due to decreased internal consistency, poor correlation between items or items-total correlation.

### Exploratory Factor Analysis
The KMO index (0.80) and the BTS ($\chi^2 = 1005.02$; $df = 15$; $p < 0.001$) supported the use of the EFA. The EFA showed two factors, with a balanced factor structure, that explained 79.36% of the variance. As expected, the two factors found correspond to the neglect (Factor 1) and the boredom (Factor 2) dimensions (**Table 1**), by confirming H1.

## Phase 2: Validity, Reliability, and Invariance of the Scale
### Confirmatory Factor Analysis
When comparing the two competing models, the bidimensional model showed the best (excellent) fit indices (**Figure 1**), by

**TABLE 2 |** Results of the Mediation Analysis of Intention as a Mediator in the Relationships Between Neglect and Protective Behavior (Hypothesis 5a), Between Boredom and Protective Behavior (Hypothesis 5b), Between Social Norms and Protective Behavior (Hypothesis 5c), and Between Behavioral Control and Protective Behavior (Hypothesis 5d). Study Attitudes, behaviors, and psychological health in time of pandemic, Spain, 2021.

| | Consequent | | | |
| --- | --- | --- | --- | --- |
| | Intention (M) | | Protective Behavior (Y) | |
| **H5a** | Coeff | SE | Coeff | SE |
| Constant | 4.66*** | 0.12 | 3.44*** | 0.26 |
| Neglect (X) | −0.25* | 0.04 | −0.02 | 0.04 |
| Intention (M) | — | — | 0.54*** | 0.05 |
| Model settings | $R^2 = 0.10$ | | $R^2 = 0.29$ | |
| | $F(1, 318) = 35.46***$ | | $F(2, 317) = 64.87***$ | |
| Indir. Cond. effect | X→M→Y | | | |
| Bootstrapp (95% CI) | −0.168 [−0.232, −0.106] | | | |
| **H5b** | | | — | |
| Constant | 4.90*** | 0.18 | 3.11*** | 0.29 |
| Boredom (X) | −0.22*** | 0.04 | 0.04 | 0.04 |
| Intention (M) | — | — | 0.57*** | 0.05 |
| Model settings | $R^2 = 0.07$ | | $R^2 = 0.29$ | |
| | $F(1, 318) = 27.20***$ | | $F(2, 317) = 65.49***$ | |
| Indir. Cond. effect | X→M→Y | | | |
| Bootstrapp (95% CI) | −0.155 [−0.221, −0.092] | | | |
| **H5c** | | | — | |
| Constant | 4.88*** | 0.16 | 3.52*** | 0.29 |
| Social Norms (X) | −0.30*** | 0.05 | −0.04 | 0.05 |
| Intention (M) | — | — | 0.53*** | 0.05 |
| Model settings | $R^2 = 0.09$ | | $R^2 = 0.29$ | |
| | $F(1, 318) = 31.07***$ | | $F(2, 317) = 65.16***$ | |
| Indir. Cond. effect | X→M→Y | | | |
| Bootstrapp (95% CI) | −0.156 [−0.219, −0.097] | | | |
| **H5d** | | | — | |
| Constant | 1.49*** | 0.24 | 2.73*** | 0.25 |
| Behavioral Control (X) | 0.56*** | 0.05 | 0.24*** | 0.06 |
| Intention (M) | — | — | 0.42*** | 0.05 |
| Model settings | $R^2 = 0.28$ | | $R^2 = 0.32$ | |
| | $F(1, 318) = 125.37***$ | | $F(2, 317) = 76.55***$ | |
| Stand. Indir. effect | X→M→Y | | | |
| Bootstrapp (95% CI) | 0.222 [0.156, 0.295] | | | |

*p <. 05; **p <. 01; ***p < 0.001.
X = Dependent variable; M = Mediator; Y = Independent variable; Stand. Indir. effect. = Completely standardized indirect effect; CI = Confidence interval; Coefficient = coefficient; SE = standard error.

confirming H1 (despite this, the single-factor model had acceptable fit indices).

### External Validity
#### -Correlation Analyses.
Both the PFS and its neglect dimension correlated as expected with all the studied variables; in contrast, the boredom dimension was 1) significantly correlated only with the social norm perception and the intention to behave in a protective way, and 2) marginally correlated with security social values orientations and PB related to the pandemic. No relation was found between the boredom dimension and the conformism social value orientation

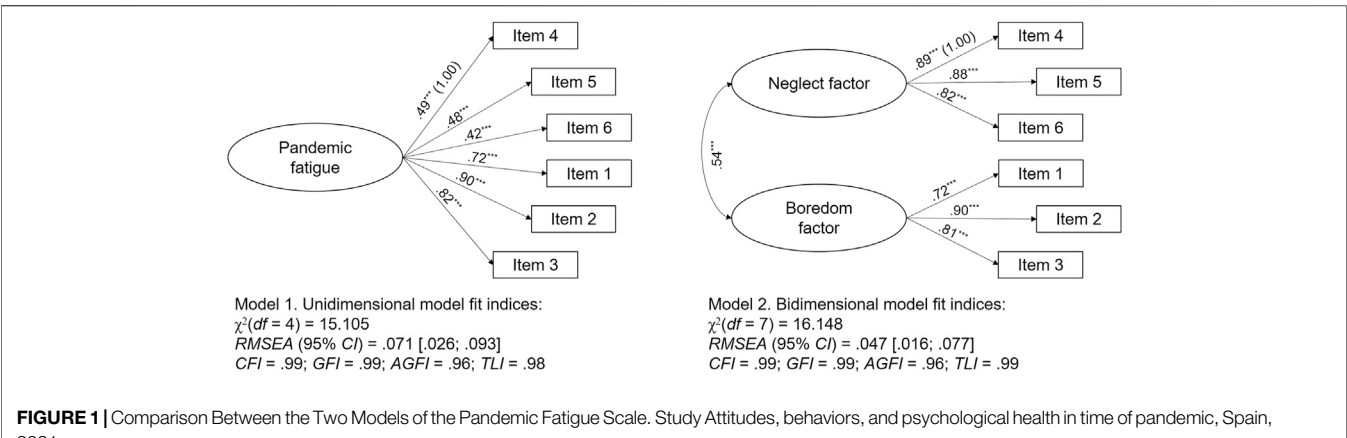

**FIGURE 1 |** Comparison Between the Two Models of the Pandemic Fatigue Scale. Study Attitudes, behaviors, and psychological health in time of pandemic, Spain, 2021.

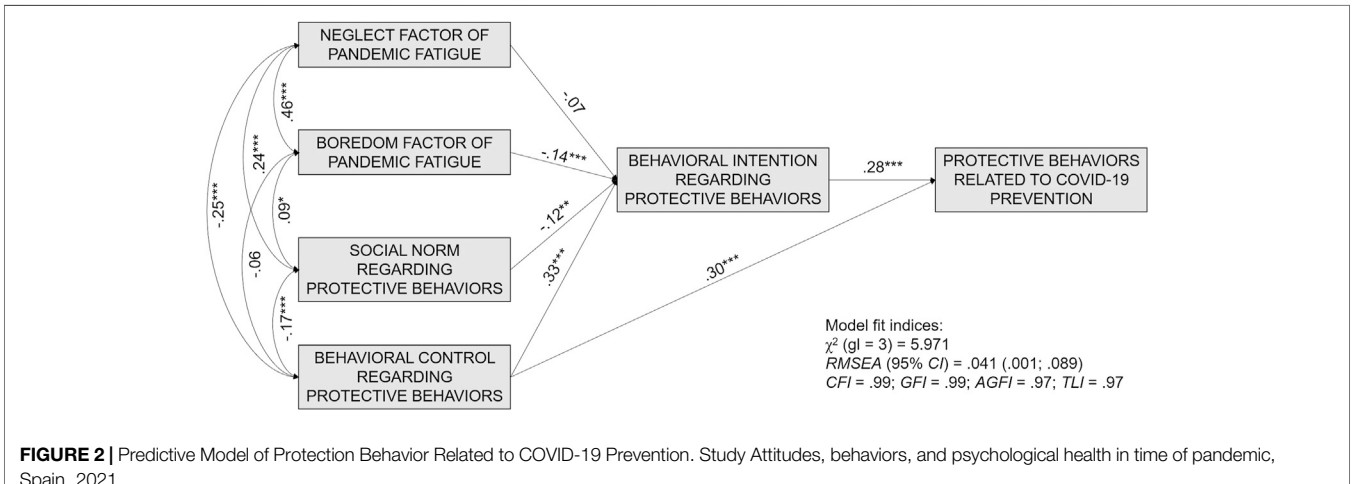

**FIGURE 2 |** Predictive Model of Protection Behavior Related to COVID-19 Prevention. Study Attitudes, behaviors, and psychological health in time of pandemic, Spain, 2021.

and the perceived behavioral control (**Supplementary Table 3**). H2 and H3 were confirmed, and H4 was partially confirmed.

*-Mediation Analysis.*
The results of the mediation analysis (**Table 2**) revealed that intention mediated the relationships between 1) neglect and PB, 2) boredom and PB, 3) social norms and PB, and 4) behavioral control and PB, thus confirming all the mediational hypotheses (H5a to H5d).

*-Predictive Model of Protective Behaviors.*
The fit indices of the predictive model of PB were adequate (**Figure 2**), confirming the relationships explored in the previous mediation analyses, as well as the study hypotheses. The independent variables boredom attitude and behavioral control were predictors of PB, with boredom being an indirect predictor mediated by the intention variable and behavioral control being both a direct and an indirect predictor again mediated by the intention variable.

## Analysis by Gender

Fit indices (**Table 3**) for the male and the female samples were adequate, and when performing the multigroup analyses, no significant differences were found between the unconstrained and the fully constrained models. Then, the model was invariant for gender, being valid for both men and women.

The analyses of mean differences (**Figure 3**) by gender (ANOVA) showed significant differences between men and women in the PFS ($F$ (1, 594) = 5.51, $p$ < 0.02) and the neglect factor ($F$ (1; 594) = 6.48, $p$ < 0.02). However, no differences were found in the boredom factor ($F$ (1; 594) = 2.27, ns). H6 was confirmed.

## DISCUSSION

Given the relevance of PF, the lack of scales to measure this concept justifies the relevance of this study. The PFS provides researchers and other professionals with a valid and reliable

**TABLE 3 |** Fit indices of the different models. Study Attitudes, behaviors, and psychological health in time of pandemic, Spain, 2021.

| Models | $\chi^2$ (df) | GFI | AGFI | CFI | TLI | RMSEA (95% CI) | Multigroup $\Delta\chi^2$ (df)/p |
|---|---|---|---|---|---|---|---|
| Values for the men sample | 10.45(7) | 0.98 | 0.95 | 0.99 | 0.99 | 0.05 [0.01,0.10] | — |
| Values for the women sample | 9.74 [7] | 0.99 | 0.97 | 0.99 | 0.99 | 0.03 [0.01,0.07] | — |
| Multigroup unconstrained model (MU) | 20.20 [14] | 0.98 | 0.96 | 0.99 | 0.99 | 0.03 [0.01,0.05] | MFC → MU |
| Multigroup fully constrained model (MFC) | 22.90 [19] | 0.98 | 0.97 | 0.99 | 0.99 | 0.02 [0.01,0.04] | 2.701 [4]/ns |

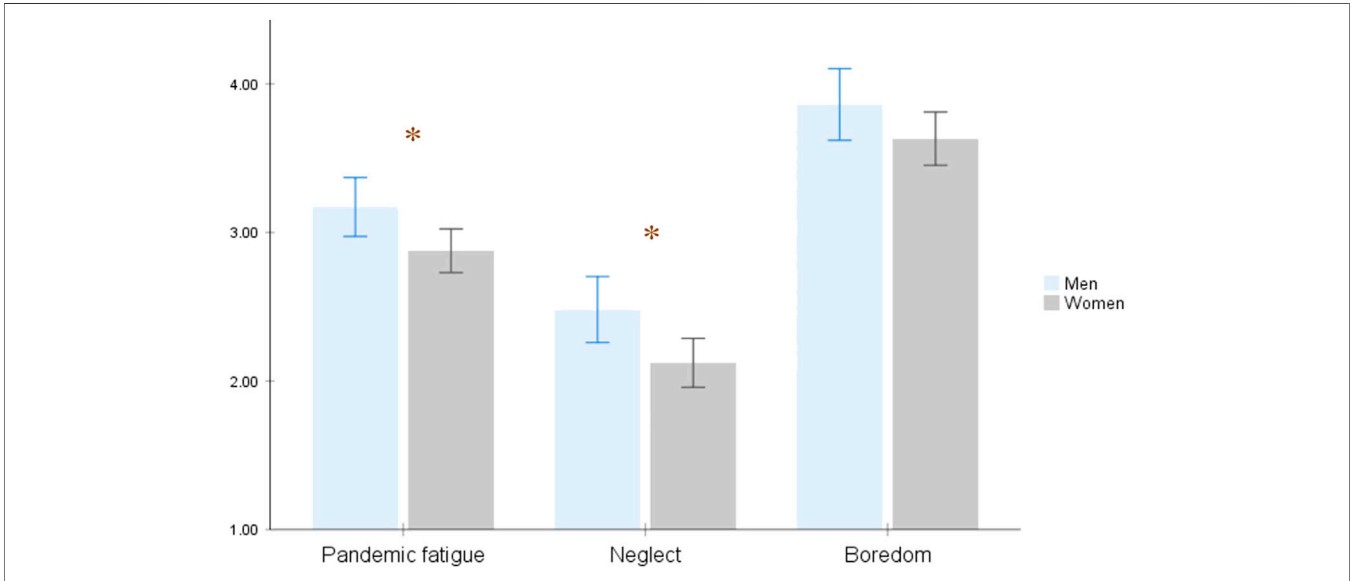

**FIGURE 3 |** Differences by Sex on the Pandemic Fatigue Construct and Its Dimensions. Study Attitudes, behaviors, and psychological health in time of pandemic, Spain, 2021.

instrument to assess the level of PF in society and to explore the potential predictors and consequences of such psychological response to the chronic pandemic situation, and then to propose practical interventions to mitigate the negative effect of PF on PB, and in turn, indirectly, on the spread of COVID-19.

# The Pandemic Fatigue Scale, a Bifactorial, Reliable, Valid, and Invariant Scale
## Bidimensionality Structure of the Scale
The bidimensionality of the PFS was confirmed, with a robust adjustment for the bifactorial structure. The two identified dimensions correspond to the expected one, on the basis of the definition of PF as a natural response to chronic, invasive restrictions implemented to minimize the effect of COVID-19 on public health and to overexposure to negative information regarding the pandemic [4–6]. Thus, the first factor corresponded to the neglect dimension, meaning the feeling of exhaustion and negative attitude related to the chronic nature of the pandemic that leads individuals to a demotivation for applying PB to prevent disease transmission. And the second factor reflects the feeling of exhaustion and negative attitude related to the prolonged and repetitive overexposure to information regarding the pandemic situation. Both factors

were composed of three items, resulting in a short scale of easy and rapid application.

## External Validity of the Scale
The pattern of relation shown among subscales and between subscales and other theoretically related measures supported the validity of the bifactorial structure and provided strong evidence for scale validity.

As expected, the neglect dimension of the PFS correlated with personal values of both security and conformity. In accordance with previous studies [15], which have shown that individuals with higher conservation values adopt higher PB, the results confirmed that the more individuals are oriented toward self-protection, with high conservation values (high conformity and security), the less they feel unmotivated to follow PB as a consequence of the prolonged pandemic situation and the chronic nature of restrictions (meaning lower values for the neglect dimension of the PFS). This seems logical, because people with high personal values of conformity and security are characterized by a high orientation toward self-protection [14], and thus, to protect themselves they will continue being motivated to engage in PB even in a situation of chronic restrictions [15].

Nevertheless, the boredom factor of the PFS was only marginally related to personal values of security and not correlated with conformity. Again, this result provides more evidence for the

bidimensionality of the PFS. The boredom factor is not so related to security as the neglect scale. It represents a demotivation not to follow the PB related to COVID-19 transmission, but to follow the information related to the pandemic, as well as demotivation provoked by the prolonged overexposure of negative COVID-19 information. In this sense, if not following the PB could be perceived as a relevant problem for individual self-protection, not following the information regarding the evolving pandemic may not be perceived as a relevant problem for self-protection, and thus the boredom factor must not be related to social values oriented toward self-protection, such as security and conformity [14].

The expected relationships also emerged between the pandemic scale and its dimensions with the intention to adopt and actual adoption of PB oriented to avoid COVID-19 exposure. As expected, and considering that the PF construct is conceived as a feeling of demotivation due to the prolonged pandemic situation and its associated stricter restrictions and information overload [4], and therefore it has an inherent motivational component, the results showed that the more individuals perceived PF (in both its neglect and boredom dimensions), the less they presented intentions to behave in a protective way and the less they adopted PB against COVID-19. People who, because of the chronic nature of the crisis, feel unmotivated to follow PB (neglect dimension of the PF construct) showed lower intention to adopt PB and reported lower PB. These results are in accordance not only with the definition of the PF construct itself [4] but also with the claims of other authors [3, 7]. In the same way, people who, because of the chronic nature of the crisis, feel unmotivated to follow the pandemic-related information (boredom dimension of the PF construct) also showed lower intention to adopt PB; nevertheless, no relation was found between the boredom dimension of the PFS and the adoption of PB. This lack of relation may be due to other variables that may be more relevant and act as mediators in the behavioral process, as indicated by the TPB [16, 19, 26] and as we have seen on the mediational analyses preformed.

In a second instance, we tested the external validity of the PFS by exploring a model in which the two different PF dimensions, jointly with the perceived social norm about PB and the perceived behavioral control toward PB, could predict individual PB, mediated by the behavioral intention, as indicated by the TPB [16, 19, 26]. The mediation analyses confirmed the expected mediating role of intention to behave in a protective way against COVID-19. Moreover, the expected relations were supported in the path analysis. Therefore, both PF dimensions, as well as perceived social norms and behavioral control, predicted individual PB through the behavioral intention. Thus, these results suggest that, meanwhile—due to the chronic nature of the public health crisis—although feelings of demotivation to follow the recommended PB against COVID-19 and the information regarding the pandemic can produce a reduction in individual PB, the more direct predictive factor of COVID-19 seems to be the intention to behave in a protective way and the perception of difficulty or ease in performing these behaviors, as suggested by the TPB [16, 19, 26].

Moreover, also in accordance with the TPB [16, 19, 26], the results show that this intention to behave in a protective way is predicted not only by PF but also by the perception of what other valued individuals think and act regarding PB—meaning the perceived social norm about the adoption of COVID-19 prevention measures. Thus, even with a demotivation to perform COVID-related PB and to follow COVID-related information due to the prolonged pandemic situation, people may have high intention to behave in a protective way, maybe because they perceive that their reference group expects them to behave in a protective way (meaning that they have a high perceived social norm), or because they perceive those PB as very easy to perform. In contrast, even if people are not demotivated to perform COVID-related PB and to follow COVID-related information due to the prolonged pandemic situation, they may have no intention to behave in a protective way because they perceive that their reference group usually does not adopt these kinds of PB or because they perceive that the behaviors are very hard to adopt.

## Invariance of the Scale

As the health consequences of the disease—both mental and physical—are different for men and women [21, 32, 33], there is a need to perform COVID-19 research with a gender perspective [20]. Therefore, the invariance of the scale and its structure were tested across a multi-group analysis that confirmed the validity of the bifactorial structure for both men and women. Thus, the bifactorial PFS is a valid and invariant scale that can be used for both men and women.

As men and women have shown a different psychological adjustment to extreme situations such as the public health crisis we are experiencing, analyzing the different impact of PF on men and women throughout the pandemic's different phases seems to be essential. Our invariant scale gave us the possibility to explore this question.

As expected, the results show that women had lower levels of the neglect dimension of PF, but the same levels of boredom as men. Thus, the feeling of exhaustion suffered by women leaves them with a similar general boredom and demotivation regarding COVID-19-related information overload as with men; nevertheless, these feelings of pandemic exhaustion and the chronic nature of strict prevention measures leave men with higher demotivation to follow the PB than for women. These results are congruent with previous studies that women adopt more PB related to COVID-19 prevention [22–24], and thus they support the external validity of the PFS. This gender difference in the COVID-19 PB could be explained by the differential socialization experienced by men and women, in which women are usually socialized to be more self-protection oriented [25] and more obedient and disciplined [34]. In this sense, our results support this statement, as significant differences were found between women and men in both the security and conformity values.

## Limitations and Future Research

Data collected were cross-sectional and non-cross-cultural in nature. Future research should deploy longitudinal and cross-cultural methodologies to examine the development of PF over time, and to compare the scores and relationship patterns of PF and other variable in different cultures. Another limitation of the

study is that no health or medical background data were obtained from the sample. Future research should explore the relation between PF and the health status and medical situation of the individuals, given that these variables may influence the individuals' perception about their vulnerability to COVID-19, which in turn may influence their PF.

An additional limitation of the study is related to the applicability associated with the characteristic of the construct itself. Given that PF is based on how overexposure to negative information may influence our adherence to health indications, the diffusion of information and the evolution of the pandemic in each country could be an important issue to consider when using this instrument in samples from other cultures. PF is also a very specific construct that might be not applicable in other circumstances than in pandemic situations related to COVID-19. Nonetheless, it can be highlighted that, with slight wording modifications, the scale might be useful and suitable in other epidemic or pandemic situation related to other viruses, to more general health situations, or in coping with uncertain health situations that also can produce fatigue in individuals. Finally, some additional dimensions could have been explored for the scale development, such as considering not only messaging fatigue, but also desensitization toward information during a pandemic [13] or physical fatigue. Future research could analyze the convenience of including additional dimension to the PFS, such as desensitization toward pandemic-related information as a potential demotivational factor regarding PB related to COVID-19.

## Conclusion

The analyses provide evidence for the validity and reliability of the PFS. This instrument represents a short, valid, and invariant scale to measure PF that can be used both by practitioners and researchers. In the actual crisis, in which it is fundamental to maintain PB to avoid disease transmission, the creation of such a scale is of relevance.

## ETHICS STATEMENT

Ethical review and approval was not required for the study on human participants in accordance with the local legislation and institutional requirements. The patients/participants provided their written informed consent to participate in this study. The study was conducted in accordance with the Declaration of Helsinki. Before giving their consent, participants were informed about the study objectives and the voluntary nature of their participation, that their anonymity was ensured, and that they could withdraw from the study whenever they wanted.

## AUTHOR CONTRIBUTIONS

EC has conceived and designed the study; analyzed and interpreted the data; written the paper; and supervised the study. She has full access to all the data and take responsibility to the integrity of the data and the accuracy of the data analysis. MAM has actively taken part in the redaction of the manuscript. CT has helped to conceived and design the study, to interpret the data, and has revised the manuscript. AA has helped to conceive and design the study, to collect the data, and has revised the manuscript. RC-M has helped to collect the data and has revised the manuscript. BL has helped to collect the data and has revised the manuscript.

## FUNDING

The data collection was financially supported by the University of Cordoba, in the Ucoimpulsa modality of the Own Research Plan of the University of Cordoba, in which EC is the main researcher.

## CONFLICT OF INTEREST

The authors declare that the research was conducted in the absence of any commercial or financial relationships that could be construed as a potential conflict of interest.

## SUPPLEMENTARY MATERIAL

The Supplementary Material for this article can be found online at: https://www.ssph-journal.org/articles/10.3389/ijph.2021.1604260/full#supplementary-material

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
