## [Reviewer comments · International Journal of Public Health]

Peer Review Report

Review Report on Construction and Validation of a Brief Pandemic Fatigue Scale in the Context of the Coronavirus-19 Public Health Crisis

Original Article, Int J Public Health

Reviewer: Hiram Ortega

Submitted on: 30 Jun 2021

Article DOI: 10.3389/ijph.2021.1604260

EVALUATION

Q 1 Please summarize the main findings of the study.

This study provides an instrument with good clinimetric characteristics for evaluating pandemic related issues like boredom and neglect with a good sample size and useful for general population.

Q 2 Please highlight the limitations and strengths.

Strengths: good sample size, external validation and a path analysis supported by theory of planned behavior

Limitations: cross-sectional design, mental health and medical background were not described in the sample

Q 3 Please provide your detailed review report to the authors. The editors prefer to receive your review structured in major and minor comments. Please consider in your review the methods (statistical methods valid and correctly applied (e.g. sample size, choice of test), is the study replicable based on the method description?), results, data interpretation and references. If there are any objective errors, or if the conclusions are not supported, you should detail your concerns.

Major concern:

It would be desirable to specify if any informed consent were applied to participants or if any IRB was consulted.

Minor concerns:

It would be desirable to specify phase 2 duration, it seems that phase 2 initiated the same day that phase 1 and last until Christmas.

In discussion section, at beginning, it is described that a neglect dimension of PFS is correlated with conformism and security values, and that boredom dimension is not correlated, however it seems to be a confusion regarding strength of the correlation and the "p" values, since none of these two dimensions have a strong nor moderated correlation (direct or inverse) with conformism or security values (showed in supplementary file 4)

PLEASE COMMENT

Q 4 Is the title appropriate, concise, attractive?

The title captures the utility of the instrument

Q 5 Are the keywords appropriate?

Keywords reflects the content of the study

Q 6 Is the English language of sufficient quality?

Study has an appropriate use of language

Q 7 Is the quality of the figures and tables satisfactory?

Yes.

Q 8 Does the reference list cover the relevant literature adequately and in an unbiased manner?)

Reference list includes an adequate background of the theme.

QUALITY ASSESSMENT

Q 9 Originality

Q 10 Rigor

Q 11 Significance to the field

Q 12 Interest to a general audience

Q 13 Quality of the writing

Q 14 Overall scientific quality of the study

REVISION LEVEL

Q 15 Please make a recommendation based on your comments:

Minor revisions.